# Influence of emission size distribution and nucleation on number concentrations over Greater Paris

Karine Sartelet[1], Youngseob Kim[1], Florian Couvidat[2], Maik Merkel[3], Tuukka Petäjä[4], Jean Sciare[5], and Alfred Wiedensohler[3]

[1]CEREA, École des Ponts ParisTech, EdF R&D, IPSL, 77455 Marne la Vallée, France
[2]INERIS, 60550 Verneuil en Halatte, France
[3]Leibniz Institute for Tropospheric Research, Leipzig, Germany
[4]Institute for Atmospheric and Earth System Research/Physics, University of Helsinki, 00014, Finland
[5]Laboratoire des Sciences du Climat et de l'Environnement, Gif-sur-Yvette, France

*Correspondence to:* Karine Sartelet (karine.sartelet@enpc.fr)

**Abstract.** With the growing evidence that high particle number concentrations may impact health, modelling their emissions and understanding formation processes is necessary, especially in cities where many people are exposed. As emission invento-ries of particle numbers and size distribution over cities are usually not available, a methodology is defined to estimate them from $PM_{2.5}$ emissions, ratios $PM_1/PM_{2.5}$ and $PM_{0.1}/PM_{2.5}$ by activity sector. In this methodology, a fitting parameter $\alpha_{em}$ is used to redistribute the number concentrations in the lowest emission diameter range. This parameter is chosen by comparing measured and simulated number concentrations during non-nucleation days. The emission size distribution is then finely dis-cretised by conserving both mass and number in each of the size ranges where emissions are specified. The methodology is applied over Greater Paris during the MEGAPOLI campaign (July 2009). Three-dimensional simulations are performed using the chemistry-transport model Polair3D/Polyphemus coupled to the aerosol module SSH-aerosol to represent the evolution of particles by condensation/evaporation, coagulation and nucleation, with a sectional approach for the size distribution. The model is first compared to measurements during non-nucleation days, and the influence over the month of July 2009 of three different nucleation parameterisations is assessed: a binary (sulfuric acid, water), ternary (sulfuric acid, ammonia, water) and heteromolecular (extremely-low volatile organic compounds (ELVOC) from monoterpenes and sulfuric acid). The modelled number concentrations compare very well to measurements with an average normalised mean error of 42% for the daily num-ber concentrations of particles larger than 10 nm, and 37% for the number concentrations of particles larger than 100 nm. The influence of the binary nucleation is low, and the ternary nucleation scheme leads to better simulated number concentrations (in terms of bias and error) at only one site out of three, but it systematically reduces the model to measurement correlation, suggesting that ternary nucleation may not be the dominant process in new particle formation. However, the relative bias and error, as well as the correlation at suburban sites, are systematically improved using the heteromolecular nucleation scheme involving sulfuric acid and ELVOC from monoterpenes. This suggests that heteromolecular nucleation may be important in cities, especially at suburban sites in summer, and that a better characterisation of the emissions of ELVOC precursors from traffic is needed.

# 1 Introduction

Although ongoing air-quality regulations only apply to particle mass, the number of particles may also be a hazard to human health (Win-Shwe and Fujimaki, 2011; Kelly et al., 2011; Pascal et al., 2013; Downward et al., 2018; Rivas et al., 2021). For example, Oberdörster et al. (2005); Schraufnagel (2020) showed that particulate matter (PM) of diameters lower than 100 nm ($PM_{0.1}$, also called ultra-fine particles UFP) are responsible for pulmonary inflammation. Because of their small sizes, they can tranlocate to all organs (Schraufnagel, 2020). Because the mass of UFP is negligible, it contributes little to the total mass concentration of particles, but the number concentration of UFP is high. Because current regulations govern the mass of particles of aerodynamic diameters lower than or equal to 10 $\mu$m ($PM_{10}$) and 2.5 $\mu$m ($PM_{2.5}$), UFP are not regulated by those; and differences in maps of high mass and number concentrations have been reported (Ye et al., 2020).

Although most computational fluid dynamics and chemistry-transport models have focused until very recently to accurately represent the mass of particles, modelling the number of particles has become increasingly the subject of studies. At the local scale, the number of particles was modelled at the local exhaust outlet (Albriet et al., 2010; Xu et al., 2021), in the plumes of ships (Karl et al., 2020) and at the neighbouring scale (Karl et al., 2016; Kurppa et al., 2020; Ketzel et al., 2021; Kumar et al., 2022), stressing the large influence of nucleation and primary emissions from traffic. At the regional and global scales, chemistry-transport models with model to measurement comparisons of number concentrations were performed in the United States (Jung et al., 2010; Zhang et al., 2010a; Kelly et al., 2011) and more recently in Europe (Kukkonen et al., 2016; Fountoukis et al., 2012; Patoulias et al., 2018; Fanourgakis et al., 2019; Olin et al., 2022; Patoulias and Pandis, 2022; Frohn et al., 2021). Only a few studies performed simulations over cities (Kukkonen et al., 2016; Frohn et al., 2021), with poorer statistics than at the regional/global scale (Frohn et al., 2021).

The high number concentrations are largely due to UFP (de Jesus et al., 2019). Although UFP may undergo transport and they may be formed elsewhere than the observation site (Cai et al., 2018), in cities, the high particle number concentrations are thought to mostly originate from nucleation and traffic emissions in summer (Rivas et al., 2020; Casquero-Vera et al., 2022). These particles are difficult to represent because of uncertainties in their emission, in the nucleation process, but also because of difficulties to model their growth mechanisms (Yu et al., 2019). Indeed, many of the modelling studies listed previously represent the size distribution using a log-normal approach with 3 to 4 modes (Zhang et al., 2010a; Kelly et al., 2011; Kukkonen et al., 2016; Fanourgakis et al., 2019). However, such a coarse discretisation of low-diameter particles induces large uncertainties in number concentrations (Sartelet et al., 2006; Blichner et al., 2021). Furthermore, Blichner et al. (2021) found that the aerosol number concentrations are better modelled compared to observations if a sectional scheme is used for low-diameter particles.

The emissions of particle numbers are highly uncertain, and they are usually not reported in emission inventories, such as the European emission inventory (EMEP/EEA, 2019), or city inventories. In the framework of the EUCAARI project, a number emission inventory was built over Europe by size-segregating PM mass emission for the different sectors (Kulmala et al., 2011). Most of the regional-scale studies presented above used this emission inventory (Fountoukis et al., 2012; Patoulias et al., 2018; Patoulias and Pandis, 2022) or an updated version (Olin et al., 2022). Although the number concentrations may be particu-

larly high in cities, number emissions are difficult to estimate. Kukkonen et al. (2016); Frohn et al. (2021) estimated number emissions from particle mass emissions for different anthropogenic sectors. These number emissions were then assigned to the Aitken mode of a modal size representation for 3D modelling. Such an approach is not appropriate for a sectional size representation where the aerosol dynamics is finely modelled. A methodology is needed to estimate particle number emissions and the size distribution at emissions for city-scale inventories.

Not only the primary emissions and size distribution at emission of UFP are highly uncertain, but also their formation from gas-phase precursors (nucleation), as well as the emissions of low-volatile organic vapors, which may strongly influence the growth of UFP (Patoulias and Pandis, 2022). Okuljar et al. (2021) showed that sub-3 nm particles may largely be directly emitted by traffic, but this contribution may be low during nucleation episodes. Low-volatile organic vapors are also emitted by traffic and depending on the distance from the source, they may be in gas phase (at high temperature) or in the condensed phase (after cooling to ambient temperature). The difficulty in accounting for the organic vapors in the emission inventory arises from the fact that they might already be partly included in $PM_{2.5}$ (as organic carbon, Kim et al., 2016).

Nucleation is uncertain both in terms of the gas involved and their representation. Several parameterisations of binary nucleation (involving sulfuric acid and water) or ternary nucleation (involving sulfuric acid, ammonia and water) exist. Zhang et al. (2010b) compared binary and ternary nucleation parameterisations and found differences of several orders of magnitude among the parametrised nucleation rates. Among the parameterisations tested, those with a simple power law to describe the binary nucleation of sulfuric acid (Sihto et al., 2006; Kuang et al., 2008) compared best to observed nucleation rates. Zhang et al. (2010b) reported that the commonly-used binary parameterisations of Kulmala et al. (1998); Vehkamäki et al. (2002) or the ternary parameterisations of Napari et al. (2002); Merikanto et al. (2007), which are based on classical homogeneous nucleation models, over-estimate the nucleation rate. Organic vapors, such as highly oxygenated organic molecule (HOM) may also be involved in nucleation (Tröstl et al., 2016; Sulo et al., 2021). The large influence of the heteromolecular nucleation of sulfuric acid and organics has been underlined in a global modelling study (Zhu and Penner, 2019). However, the influence of heteromolecular nucleation was not assessed at the regional or city scale, where HOM are believed to mostly contribute to the growth of nano particles (Patoulias and Pandis, 2022).

This paper aims at modelling the number of particles over Greater Paris during summer, first by defining a methodology to estimate primary number emissions, and second by estimating nucleation parameterisations that best represent measurements. The simulations are performed during the summer MEGAPOLI campaign. The first section presents the model and the measurement data. The second section defines a methodology to estimate the number emissions from the different emission sectors. Finally, the fourth section studies the influence of nucleation (binary, ternary and heteromolecular with organics) on the number concentration.

## 2 Presentation of the model and data

### 2.1 The model

Simulations are performed with the 3-dimensional (3D) chemistry-transport model Polair3D (Sartelet et al., 2007) of the Polyphemus platform, which is coupled to the aerosol module SSH-aerosol (Sartelet et al., 2020). The gas-phase chemistry model is CB05, modified to represent the formation of semi-volatil organic compounds that may condense onto particles and form secondary organic aerosols (SOA) (Kim et al., 2011; Chrit et al., 2017). The considered SOA precursors are anthropogenic (toluene, xylenes, intermediate and semi-volatile organic compounds) and biogenic (monoterpenes, sesquiterpenes, isoprene). The myriad of SOA species formed during the oxidation of those precursors is modelled with surrogate organic molecules of representative physico-chemical properties (Couvidat et al., 2012; Sartelet et al., 2020). Some of the surrogates may be considered as non-volatile: the surrogate BiA3D (3-methyl-1,2,3-butane tricarboxylic acid) from the monoterpene oxidation, the surrogates Monomer ($C_{10}H_{14}O_9$) and Dimer ($C_{19}H_{28}O_{11}$) from the monoterpene autoxidation, the surrogate AnClP from the xylenes and toluene low-NOx oxidation, the surrogate SOAlP (secondary organic aerosol of low volatility) from the oxidation of anthropogenic semi-volatile organic compounds. The growth of UFP is strongly impacted by the condensation of these low-volatility compounds as well as coagulation. Therefore, numerically, the condensation of non-volatile compounds is solved dynamically with nucleation and coagulation processes, using the ETR (explicit trapezoidale rule) numerical scheme. In each section, particles grow because of condensation, leading to variations of the section diameters. Because the bound diameters of each section should remain fixed to ensure numerical consistency with coagulation and 3D transport, the number and mass concentrations are redistributed at each time step on the the fixed size (diameter) sections using the Euler-coupled approach (Devilliers et al., 2013). The condensation/evaporation of semi-volatile compounds is computed by assuming bulk thermodynamic equilibrium between the gas and the particle phases. The condensing matter estimated from bulk equilibrium is distributed over the aerosol size distribution by using weighting factors for each size section based on their condensation/evaporation kernel of the condensation/evaporation rate.

Different parameterisations of nucleations are implemented: the binary parameterisation of Kuang et al. (2008) involving sulfuric acid and water; the ternary parameterisation of Napari et al. (2002) involving sulfuric acid, water and ammonium; the heteromolecular parameterisation of Riccobono et al. (2014) involving sulfuric acid and oxidised monoterpene compounds. Concerning this heteromolecular nucleation, Schobesberger et al. (2013) argued that stable clusters with sulfuric acid molecules may be effectively formed from highly oxidized extremely-low volatile organic compounds (ELVOCs). The less oxidized, but more abundant oxidation products may rather drive the initial growth of the clusters. Hence, the concentration of the oxidised biogenic compounds is assumed to be equal to the concentration of ELVOCs, which are formed in the model from the autoxidation of monoterpenes (Ehn et al., 2014; Chrit et al., 2017). Several studies rescaled the ternary parameterisation of Napari et al. (2002) using scaling factors of the order of $10^{-5}$-$10^{-6}$, because of too high nucleation rates (Fountoukis et al., 2012; Patoulias et al., 2018). A scaling factor of 0.001 is used here. As the heteromolecular parameterisation of Riccobono et al. (2014) also led to too high number concentrations, it is rescaled by a factor 0.1.

## 2.2 Simulation setup

The simulation domain (see Figure 4) and the model input data (meteorology and boundary conditions) are the same as in Royer et al. (2011); Couvidat et al. (2013). Only 5 size sections between 0.01 $\mu$m and 10 $\mu$m were used in these studies. To represent the aerosol dynamics, including the nucleation process, the discretisation of particle diameters starts at 1 nm here, and the number of sections is increased to 25. The bound diameters of the sections used in the modelling are (in $\mu$m): 0.001, 0.00133,0.00177, 0.00237, 0.00316, 0.00421, 0.00562, 0.00750, 0.01, 0.0141,0.0199 0.0282 0.0398,0.0562, 0.0794, 0.112, 0.1585, 0.224, 0.316, 0.447, 0.631, 0.891, 1.26, 2.5, 5.0, 10. The distribution of boundary conditions and emissions into 25 size sections is done offline, prior to the simulation, using the algorithm detailed in Appendix. Because the larger scale simulations from the nesting domain presented in Royer et al. (2011); Couvidat et al. (2013) did not include particles of diameters lower than 0.01 $\mu$m, the boundary conditions for particles between 0.001 and 0.01 $\mu$m are fixed to 0. The allocation of emissions to the different sections is detailed in the following section.

## 2.3 Size distribution at emission

Anthropogenic emissions are obtained from the Airparif 2005 inventory, which provides emissions for the different category sectors, defined by Selected Nomenclature for Air Pollution (SNAP). For particulate emissions, only $PM_{10}$ and $PM_{2.5}$ emissions are available. Emissions of PM from traffic are assumed to be made of 50 % elementary carbon, 40 % organics, and 10 % non-volatile noncarbonaceous PM (Couvidat et al., 2013). Emissions of intermediate, semi and low volatile organic compounds (IVOC, SVOC, LVOC respectively) are estimated from the organic mass of $PM_{2.5}$ as detailed in Sartelet et al. (2018): the mass of organic vapors is estimated by multiplying by 1.5 the organic mass (Kim et al., 2016). The emitted organics are then divided into volatility classes: 25 % is assigned to LVOC, 32 % to SVOC and 43 % to IVOC (Couvidat et al., 2012). To determine number emissions, the size distribution of $PM_{2.5}$ is estimated using ratios of $PM_1/PM_{2.5}$ and $PM_1/PM_{0.1}$ from the UK National Atmospheric Emissions Inventory (NAEI) for each activity sector. These factors are presented in Table A2 of Appendix A. To represent $PM_{0.1}$, $PM_1$ and $PM_{2.5}$ emissions, the size range of diameters between 0.01 $\mu$m and 10 $\mu$m is divided into 5 sections regularly distributed in log-space, and of bound diameters (in $\mu$m) 0.01, 0.0398, 0.1585, 0.631, 2.11, 10. Emissions of coarse particles ($PM_{10}$-$PM_{2.5}$) are assigned to the section of diameters between 2.11 and 10 $\mu$m. Emissions of fine particles ($PM_{2.5}$-$PM_1$) are assigned to the section of diameters between 0.631 and 2.11 $\mu$m. Note that 0.631 and 2.11 $\mu$m are used as bound diameters for $PM_1$ and $PM_{2.5}$, because PM is defined for aerodynamic diameters, while the model uses the diameter of spherical particles. Aerodynamic diameters of 1 $\mu$m and 2.5 $\mu$m correspond to diameters of 0.631 and 2.11, assuming a particle density of 1.58 g.cm$^{-3}$ and approximating the Cunningham correction factor following DeCarlo et al. (2004); Jung et al. (2020). Emissions of $PM_{0.1}$ are assigned in the size range of diameters between 0.01 $\mu$m and 0.1585 $\mu$m. The bound diameter of 0.1585 $\mu$m is reasonable for $PM_{0.1}$ at emissions, because particles may then be irregular with a diameter larger than the aerodynamic diameter (DeCarlo et al., 2004). However, the mass of UFP particles $PM_{0.1}$ is redistributed arbitrarily between the low range of UFP diameters (between 0.01 $\mu$m and 0.0398 $u$m) and the high range (above 0.0398 $\mu$m) using a distribution coefficient $\alpha_{em}$, i.e. an emission ratio ($\alpha_{em}$, (1-$\alpha_{em}$)) distributes $PM_{0.1}$ in 2 size ranges. To determine this arbitrary

distribution coefficient, simulations are compared to measurements during non-nucleation days using three different value of $\alpha_{em}$: 10%, 15% and 25% (section 3). The mass allocated to the section between 0.01 $\mu$m and 0.0398 $\mu$ corresponds to $\alpha_{em}$ times the mass of $PM_{0.1}$, and the mass allocated to the section between 0.0398 $\mu$ and 0.1 $\mu$m corresponds to (1-$\alpha_{em}$) times the mass of $PM_{0.1}$. Note that particles of diameters lower than 0.01 $\mu$m are not emitted here, although diesel vehicles may emit

such small particles (Kuuluvainen et al., 2020). However, the work of Olin et al. (2022) suggests that these emissions may not strongly affect the number concentrations at background sites, because of the coagulation of emitted particles.

     Although 5 size sections are defined for emissions, as much as 25 size sections are used in the model to represent the aerosol dynamic. To specify emissions in the range of the size sections of the model, the size distribution at emission is progressively refined by dividing each of the size section at emission into two smaller size sections, keeping both the emitted mass and

number concentrations constant during each division. The algorithm used for this division is detailed in Appendix A.

## 2.4    Measurements

Concerning number concentrations, measurements were performed between 1 and 31 July 2009 at three sites: the LHVP site, a background urban site in the center of Paris, the SIRTA site, a background suburban site in the south west of Paris, and the GOLF site, a background suburban site close to a golf course in the north east of Paris (see Figure 4). At the LHVP site, the

number concentrations were monitored using a Twin Differential Mobility Particle Sizer (TDMPS) for diameters between 3 and 635 nm. An Air Ion Spectrometer (AIS) monitored the size distribution of ambient (not dried) positive and negative air ions of mobility diameters ranging from 0.8 to 40 nm. At the GOLF site, number concentrations were monitored using an Electrical Aerosol Spectrometer (EAS) for diameters between 3 nm and 10 000 nm. At the SIRTA site, number concentrations were monitored using a Scanning Mobility Particle Sizer (SMPS) for diameters between 10 and 500 nm. The monthly-average size

distribution is plotted in Figures 7, 8 and 9 at the SIRTA, LHVP and GOLF sites respectively. The lowest number concentrations are measured at the suburban SIRTA site. The highest number concentrations are measured at the suburban GOLF site. Further details about the measurements performed may be found in Pikridas et al. (2015). Mass concentrations were also monitored, allowing to validate the modelled particle mass concentrations. The mass concentrations of sulfate, nitrate, ammonium and organics in $PM_1$ were monitored with Aerosol Mass Spectrometers (Freutel et al., 2013).

**3    Selection of non-nucleation days to determine the size distribution at emission**

The undetermined number distribution coefficient $\alpha_{em}$ at emission, defined in section 2.3, is estimated by comparing the model to the measurements during non-nucleation days at LHVP, where measurements of UFP from diameters as low as 0.8 nm are available from the AIS.

     Three distribution coefficients $\alpha_{em}$ are tested: 10%, 15%, 25% corresponding to three sets of emission ratio: 10-90%, 15-

85% and 25-75% respectively. The differences between the size distribution at emission for these sets are shown in Figure 1. The number of UFP at emission is smaller in the set 10-90% than in the set 15-85%, which is itself smaller than in the set 25-75%. The number of particles of diameters lower than 5 nm, as measured with the AIS and with the TDMPS, is shown in

Figure 2. It is the lowest at the beginning of July: it is lower than 20 cm$^{-3}$ for the 2$^{nd}$ and 3$^{rd}$ July and for the first 6 hours of the 4$^{th}$ July. This period is then selected to determine the number distribution coefficient at emission. Over that period, the averaged number concentration measured with the TDMPS consists mostly of particles of diameter larger than 10 nm (N$_{>10}$). Figure 3 compares the average number concentrations simulated over that period to the TDMPS measurements, using the different emission distribution coefficients $\alpha_{em}$: 10%, 15% and 25%. The measured N$_{>10}$ is 8216 cm$^{-3}$, while the averaged simulated N$_{>10}$ is 6172 cm$^{-3}$ for $\alpha_{em}$ =10%, 8152 cm$^{-3}$ for $\alpha_{em}$ =15% and 12883 cm$^{-3}$ for $\alpha_{em}$ =25%. Clearly, the number of UFP is too high using $\alpha_{em}$ =25%. Particles of diameters below 0.03$\mu$m are also over-estimated using $\alpha_{em}$ =15%, while they are well modelled using $\alpha_{em}$ =10%.

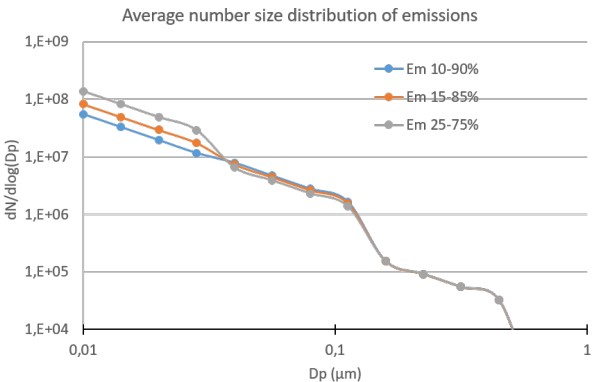

**Figure 1.** Number of emitted particles as a function of diameters in # m$^{-2}$ s$^{-1}$.

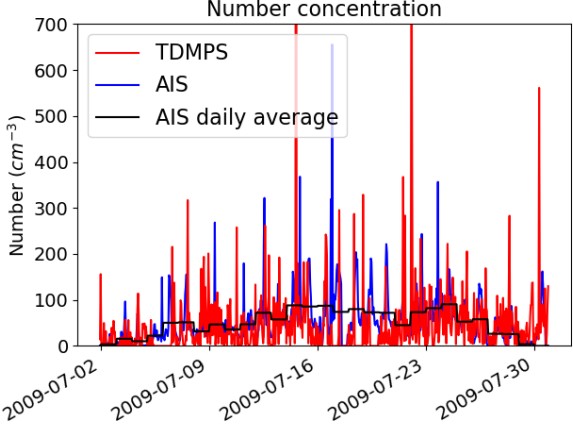

**Figure 2.** Number of particles of diameters lower than 5 nm, as measured with the AIS (in blue) and with the TDMPS (in red). The black lines represents the average daily number concentrations measured with the AIS.

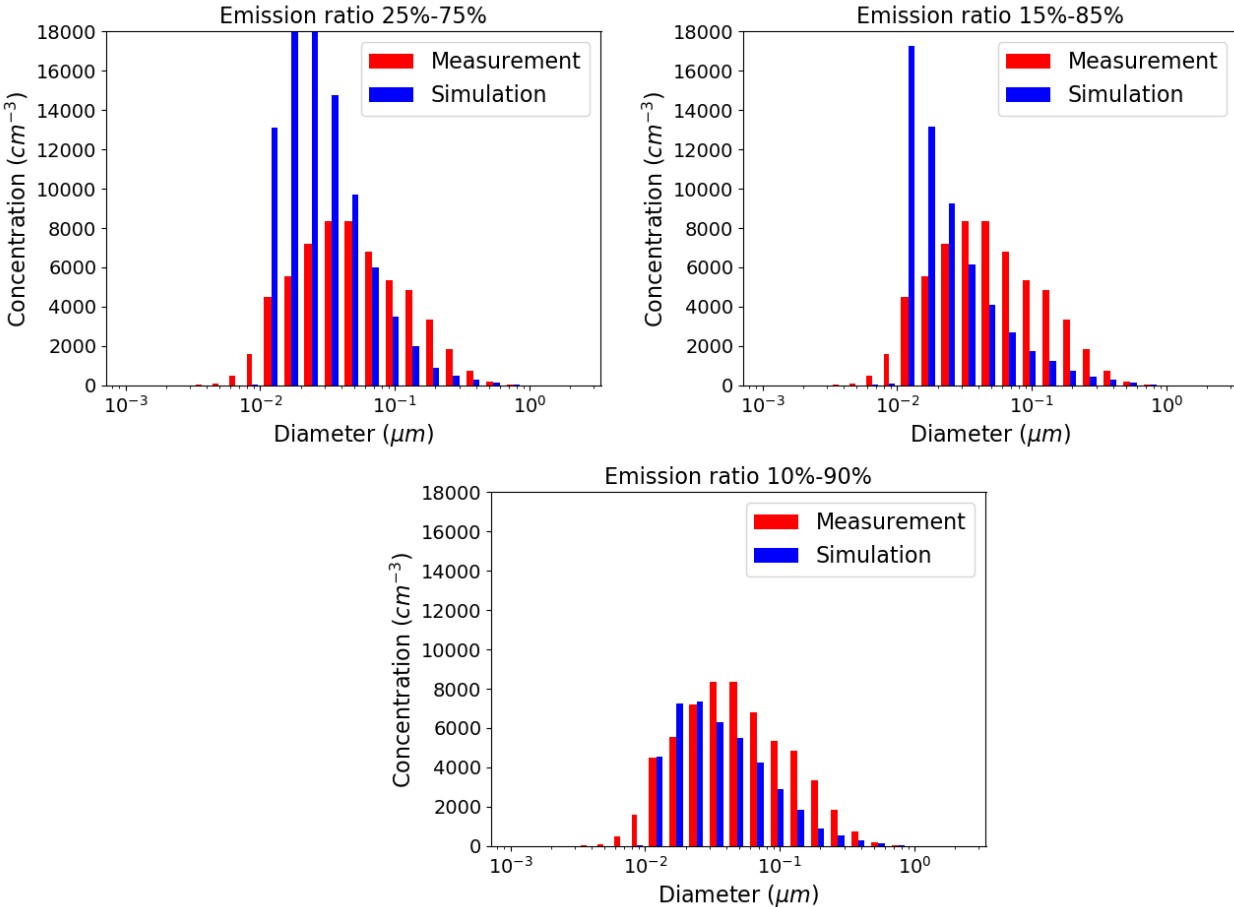

**Figure 3.** Average number concentration between the $2^{nd}$ July 00:00 and the $4^{th}$ July 6 am, simulated with an emission ratio of 25%-75% (upper left panel), 15%-85% (upper right panel), 10%-90% (lower panel). The simulated concentrations are in blue, while the concentrations measured with the TDMPS are in red.

## 4 Influence of nucleation

Although nucleation has a low influence on the mass concentrations of $PM_1$, $PM_{2.5}$ and $PM_{10}$, it has a major influence on the number concentration during July. As shown in the left panel of Figure 4, if nucleation is not taken into account, high number concentrations are observed along the main roads and motorways, and concentrations are higher near the central part of the city than in the suburbs. The main roads are clearly distinguishable on the map of number concentrations, highlighting the strong impact of traffic emissions on the simulated concentrations.

To assess the influence of nucleation, simulations are performed with different nucleation parameterisations and compared to the simulation where nucleation is ignored. The nucleation parameterisations are those detailed in section 2.1: the binary

nucleation parameterisation of Kuang et al. (2008) involving sulfuric acid and water, the ternary nucleation parameterisation of Napari et al. (2002) involving ammonia, sulfuric acid and water, and the heteromolecular nucleation parameterisation involving organic and sulfuric acid of Riccobono et al. (2014).

Nucleation leads to a large increase of number concentrations. In average over the whole domain and over the month of July, the binary, ternary and heteromolecular nucleation parameterisations lead to an increase by a factor 1.1, 2.1 and 2.8 respectively.

Figure 5 shows the relative differences between simulations taking into account one nucleation scheme and the simulation without nucleation. The increase of the number concentration by the binary parameterisation is very localised, mostly near central Paris. The increase of the number concentration by the ternary parameterisation is larger than by the binary, but it is as well very localised near Paris and its suburbs, and near large factories. The increase of the number concentration by the heteromolecular parameterisation is the largest of the three parameterisations. Although the average increase over the domain is of the same order of magnitude than the increase due to ternary nucleation, it is less localised and more homogeneously distributed. However, as shown in the right panel of Figure 4, the highest number concentrations are simulated near the central part of Paris and along the main roads and motorways, even when the heteromolecular nucleation parameterisation is used.

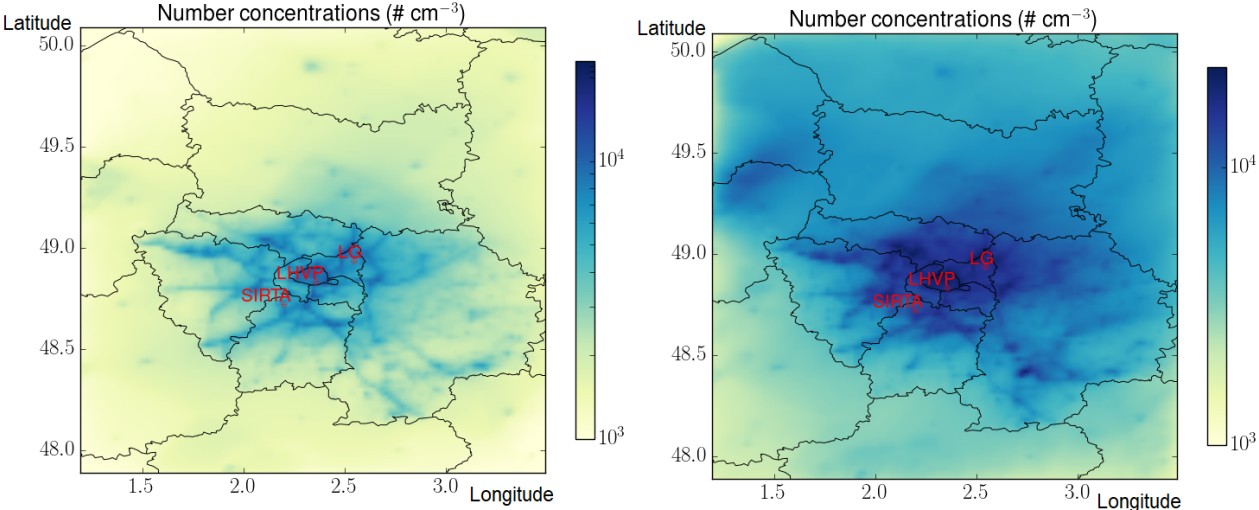

**Figure 4.** Number concentrations in $\#$ cm$^{-3}$ for the simulation without nucleation (left panel) and the simulation with heteromolecular nucleation (right panel).

## 5 Model evaluation

The simulated concentrations are evaluated using the measurements performed at the SIRTA, LHVP and GOLF sites.

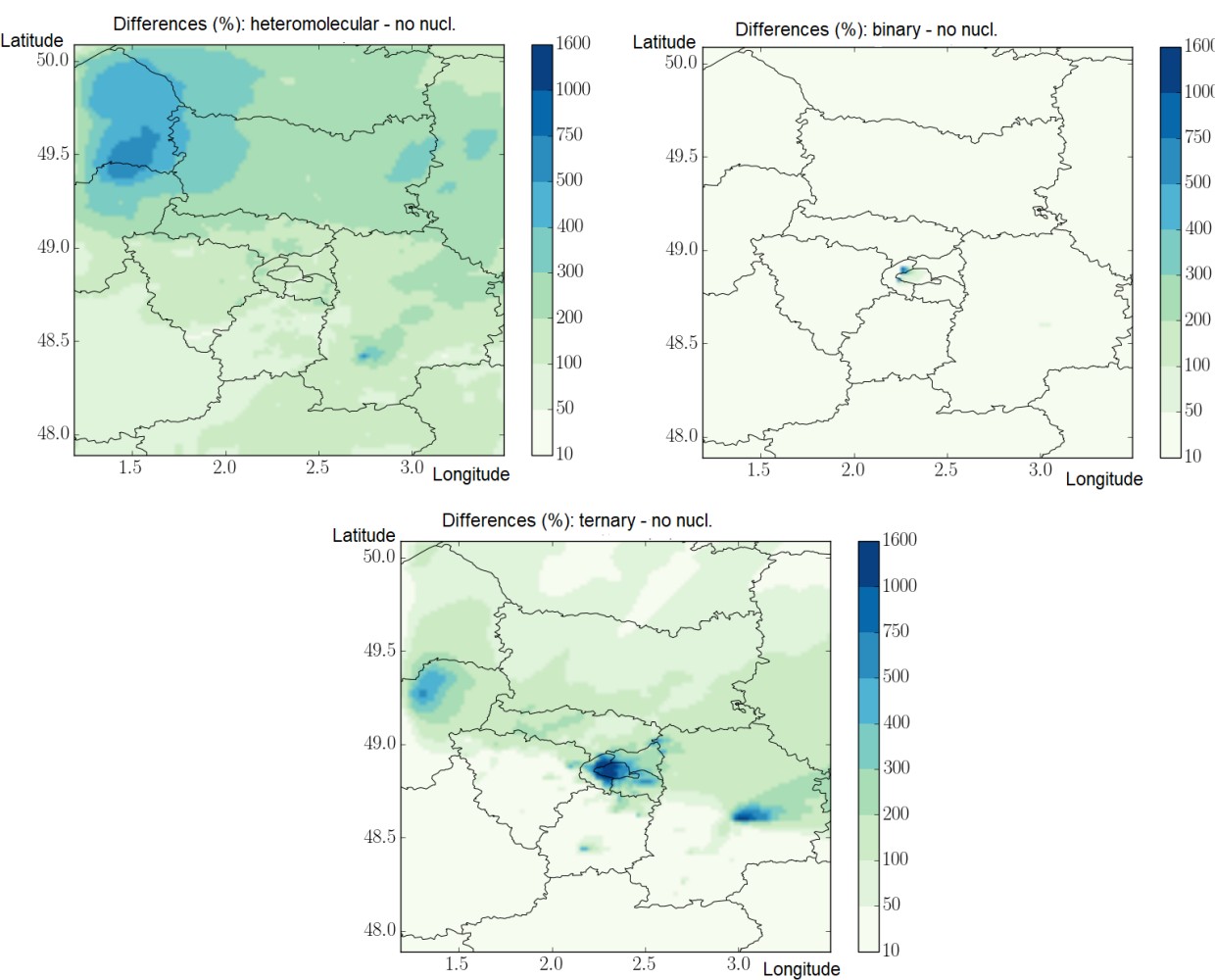

**Figure 5.** Differences in number concentrations in % between simulations with nucleation and without. The nucleation parameterisations used are respectively the heteromolecular on the top left panel, the binary in the top right panel and the ternary in the bottom panel.

## 5.1 Statistics

### 5.1.1 Mass concentration

The mass concentration of the particles and of the different compounds of particles are fairly well modelled, as shown in Table 1. $PM_{2.5}$ and $NO_2$ modelled concentrations are compared to measurements routinely carried out at background sites by the air-quality agency Airparif. For $PM_{2.5}$, as well as for sulfate, ammonium and organics in $PM_1$, the simulated concentrations satisfy the most strict performance criteria of Boylan and Russell (2006) (mean fractional error MFE below 50% and mean fractional bias MFB below 30%). The MFE for nitrate concentration is higher (88%), but the nitrate concentrations are very

low (0.2 $\mu$g m$^{-3}$) in both the measurements and the simulation. Except for nitrate, the correlations R between simulation and measurements are also high (between 57% and 76%).

Note that the statistics shown in Table 1 are not influenced by the nucleation scheme.

No measurement of NH$_3$ concentration was available in 2009. However, the simulated concentration range is realistic (see
Figure 6), considering that measurements of NH$_3$ performed recently in the Paris city center with a mini-DOAS indicate a mean concentration of 2 $\mu$g m$^{-3}$ (Viatte et al., 2021).

**Table 1.** Model to measurement comparisons of daily mass concentrations in July 2009.

|  | NO$_2$ | PM$_{2.5}$ | Sulfate$_1$ | Nitrate$_1$ | Ammonium$_1$ | Organics$_1$ |
|---|---|---|---|---|---|---|
| Number of stations | 46 | 5 | 3 | 3 | 3 | 3 |
| Meas. mean ($\mu$g m$^{-3}$) | 15.8 | 10.3 | 1.0 | 0.2 | 0.4 | 2.9 |
| Sim. mean ($\mu$g m$^{-3}$) | 14.1 | 8.2 | 0.9 | 0.2 | 0.4 | 2.4 |
| MFB (%) | -21 | -19 | -14 | -20 | 3 | -5 |
| MFE (%) | 41 | 33 | 34 | 88 | 29 | 36 |
| Correlation (%) | 73 | 76 | 69 | 32 | 74 | 57 |

### 5.1.2 Number concentration

The simulated number concentrations of particles of diameter larger than 10 nm (N$_{>10}$) and 100 nm (N$_{>100}$) are compared to the observations at SIRTA, LHVP and GOLF in Table 2. The simulation without nucleation strongly underestimates the number
concentrations N$_{>10}$ and N$_{>100}$ in July 2009 at all sites (SIRTA, LHVP and GOLF). There are no established criteria for determining how well a simulation performs against the measurement. The normalised mean bias (NMB) and the normalised mean error (NME) are often used (Fanourgakis et al., 2019; Olin et al., 2022; Patoulias and Pandis, 2022; Frohn et al., 2021). For a month of spring/summer over Europe, in Olin et al. (2022) the NME was 94% for N$_{>10}$ and 49% for N$_{>100}$ in average over 6 stations; in Patoulias and Pandis (2022) the NME was 63% for N$_{>10}$ and 45% for N$_{>100}$ on average over 26 stations.
The NMB of N$_{>30}$ ranged between 117% and 161% in Frohn et al. (2021), who used a modal approach for the size distribution. Simulations over cities led to higher errors: only monthly-means are compared in Kukkonen et al. (2016) and the NMB range between 218% and 285% for N$_{>30}$ in Frohn et al. (2021). The simulations without nucleation presented here lie in the range of errors obtained in previous studies at the European scale with NMEs between 36% and 79% for N$_{>10}$ and between 47% and 50% for N$_{>100}$. Theses statistics are improved when nucleation is used, depending on the nucleation scheme used. Using the
heteromolecular scheme for nucleation leads to very good model to measurement comparisons: the NME is 42% in average for N$_{>10}$ and 37% for N$_{>100}$.

The effect of the binary nucleation is very low, especially for $N_{>100}$ at all sites. There is almost no change in the statistics at the SIRTA suburban site for both $N_{>100}$ and $N_{>10}$. The effect of nucleation is the largest at the site in Paris center (LHVP), where it leads to an increase of the number of particules and reduces the bias from -54% to -42%. However the error slightly

increases and the correlation stronly decreases (fom 68% to 26%), suggesting that the binary nucleation scheme does not improve the model to measurement comparisons overall. At all sites, the number concentration is strongly underestimated if only binary nucleation is taken into account.

The effect of the ternary nucleation is higher than the effect of the binary one. For $N_{>100}$, the number concentration increases leading to a decrease of the bias, as well as a decrease of the error. This decrease is the strongest at the GOLF site, reducing the bias from -50% to -41% and the error from 50% to 44%. However, the model to measurement correlation decreases strongly at the GOLF site from 50% to 37%. Although the correlation also decreases because of ternary nucleation at the LHVP site, it increases from 51% to 59% at the SIRTA site. For $N_{>10}$, The increase of number concentration leads to a decrease of the bias and error at both the SIRTA and GOLF sites. As for $N_{>100}$, this decrease is the strongest at the GOLF site where the bias is reduced from -79% to -24% and the error is reduced from 79% to 51%. At the LHVP site in central Paris, the increase in $N_{>10}$ is too strong, the absolute value of the bias is not much modified: it varies from -54% to 55%, but the error increases from 55% to 97%. At all three sites, the correlation strongly decreases. The effect of the parameterisation is not completely satisfactory, because although the measurement-model bias decreases at most sites, the nucleation is too strong at the site located in the center of Paris and the measurement-model correlations decrease strongly.

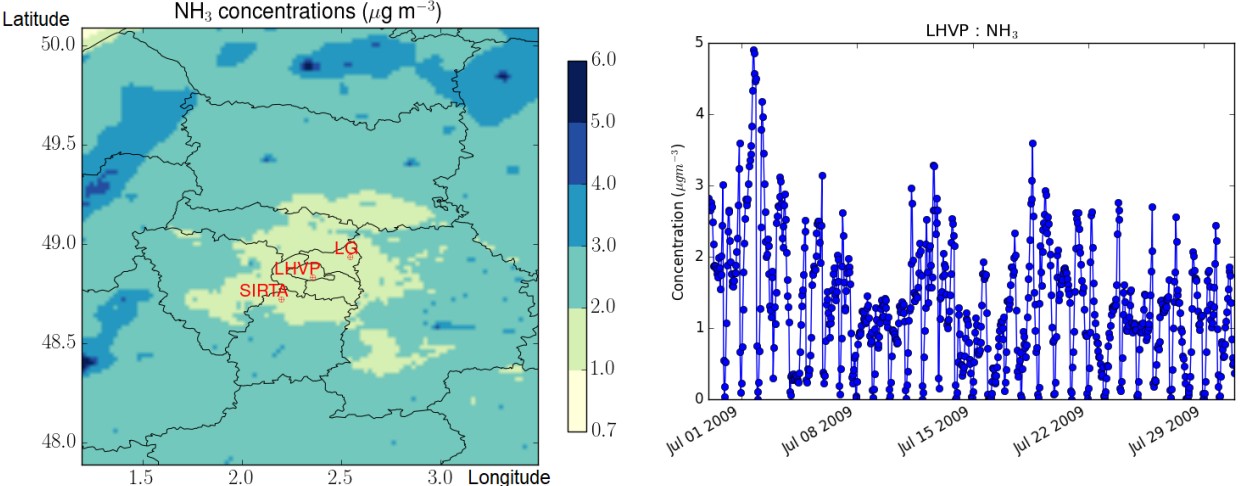

**Figure 6.** Monthly-average NH$_3$ concentration (left panel), and hourly evolution of the NH$_3$ concentrations in $\mu$g m$^{-3}$ at the LHVP site (right panel).

The effect of the heteromolecular nucleation strongly improves the bias and error of $N_{>100}$ and $N_{>10}$ at all sites. For example, at the SIRTA site, for $N_{>10}$, the bias is reduced from -54% to -17% and the error from 54% to 36%; for $N_{>100}$, the bias is reduced from -44% to -28% and the error from 47% to 36%. For $N_{>10}$, the correlation is not much modified, except at the LHVP site where it is reduced from 68% to 49%. However the correlation decreases less than with the other nucleation schemes (it decreases to 26% for the binary nucleation and to -20% for the ternary nucleation). For $N_{>100}$, the correlation also slightly decreases at the LHVP site from 77% to 68%, but it is greatly improved at both the SIRTA site (from 50% to 65%) and

the GOLF site (from 51% to 63%). These comparisons suggest that the heteromolecular nucleation improves the modelling of $N_{>10}$ and $N_{>100}$ at the suburban sites. Even though the bias and error are also improved at the city center site, the decrease of the correlation when the nucleation parameterisations are used may indicate the need to better characterize new particle formation in cities and close to traffic sites.

**Table 2.** Model to measurement comparisons of daily number concentrations of particles of diameter higher than 10 nm in July 2009 at LHVP, SIRTA and GOLF. $\bar{o}$ and $\bar{s}$ stand for mean observation and simulated concentrations respectively; Corr. stands for correlation; NMB and NME stand for normalised mean bias and normalised mean error respectively.

| | $N_{>10}$ | | | | | $N_{>100}$ | | | | |
|---|---|---|---|---|---|---|---|---|---|---|
| | $\bar{o}$ | $\bar{s}$ | Corr. | NMB | NME | $\bar{o}$ | $\bar{s}$ | Corr. | NMB | NME |
| Station | SIRTA | | | | | | | | | |
| No nucl. | 5328 | 2458 | 48 | -54 | 54 | 946 | 532 | 51 | -44 | 47 |
| Binary | 5328 | 2461 | 47 | -54 | 54 | 946 | 531 | 51 | -44 | 47 |
| Ternary | 5328 | 3241 | 27 | -39 | 51 | 946 | 571 | 59 | -40 | 43 |
| Heteromolecular | 5328 | 4396 | 49 | -17 | 36 | 946 | 680 | 65 | -28 | 36 |
| Station | LHVP | | | | | | | | | |
| No nucl. | 9852 | 4567 | 68 | -54 | 55 | 1191 | 667 | 77 | -44 | 47 |
| Binary | 9852 | 5709 | 26 | -42 | 57 | 1191 | 660 | 76 | -44 | 47 |
| Ternary | 9852 | 15302 | -20 | 55 | 97 | 1191 | 713 | 68 | -40 | 44 |
| Heteromolecular | 9852 | 7341 | 49 | -25 | 37 | 1191 | 805 | 68 | -32 | 38 |
| Station | GOLF | | | | | | | | | |
| No nucl. | 12957 | 2279 | 66 | -79 | 79 | 1221 | 615 | 50 | -50 | 50 |
| Binary | 12957 | 3195 | 23 | -75 | 75 | 1221 | 625 | 47 | -49 | 49 |
| Ternary | 12957 | 9739 | -51 | -24 | 51 | 1221 | 716 | 37 | -41 | 44 |
| Heteromolecular | 12957 | 5814 | 64 | -55 | 55 | 1221 | 808 | 63 | -33 | 37 |

## 5.2 Size distribution and hourly evolution

The strong influence of nucleation is also evident by looking at the particle size distributions averaged over the month of July in Figure 7 at the SIRTA site, Figure 8 at the LHVP site, and Figure 9 at the GOLF site. The influence of the different nucleation parameterisations is now compared. To gain more insights on the influence of the nucleation parameterisation on the size distribution and on the number formation rate, the simulated hourly evolution of the number of particles of diameters between 10 and 100 $\mu$m ($N_{10-100}$) is plotted in Figure 10 and compared to measurements. At all sites, the number concentration is strongly underestimated if only binary nucleation is taken into account, as shown by the hourly evolution of $N_{10-100}$ (Figure 10). Although a few nucleation peaks, associated with a sudden increase of $N_{10-100}$ concentrations, are sometimes simulated, they do not appear to be temporally related to those observed.

At the SIRTA site, Figure 7 shows that the influence of binary nucleation is very low. The influence of ternary nucleation is higher, but the number of particles of diameters between 20 nm and 200 nm is strongly underestimated. This number is better modelled using the heteromolecular nucleation parameterisation, although this parameterisation seems to slightly overestimate the number of particles between 10 and 20 nm. However, Figure 10 shows that the heteromolecular nucleation allows a fairly

good representation of the nucleation events associated to the different high $N_{10-100}$ concentrations, such as between 23 and 30 July.

At the LHVP site, Figure 8 shows that the influence of the binary nucleation is higher than at the SIRTA site, and leads to good model to measurement comparisons, although the number of particles of diameters between 20 and 200 nm are under-estimated. This under-estimation is less important using the heteromolecular parameterisation, which overestimate the number of particles between 3 and 10 nm. However, the hourly variations of $N_{10-100}$ concentrations are underestimated between 9 and 16 July and between 23 and 30 July when nucleation is the strongest at the LHVP site (Figure 10). This could be due to too low condensable vapors.

The ternary nucleation parameterisation performs well for number of particles of diameters above 20 nm, but the number of particles of diameter below 20 nm is strongly over-estimated (by a factor larger than 100). At the LHVP site, $NH_3$ concentrations vary between 0 and 3 $\mu g\ m^{-3}$ during most of July. As traffic is the main source of $NH_3$ in the center of Paris in July, the peaks of $NH_3$ concentrations are strongly related to the traffic peaks (Figure 6). When ternary nucleation is taken into account, peaks of particle number concentrations are simulated when $NH_3$ concentrations are high, and number concentrations are very low when $NH_3$ concentrations are low (Figure 10). These high peaks and variations of number concentrations are not observed in the measurements, suggesting that the ternary nucleation parameterisation does not perform well at the LHVP urban site.

At the GOLF site, Figure 8 shows that the number concentration is strongly underestimated if nucleation is not taken into account for particles of all diameters. The effect of binary nucleation is very small. The heteromolecular nucleation parameterisation perfoms better. Although it leads to acceptable statistics, the number concentration is still under-estimated for particles of all diameters. A good representation of the size distribution is obtained using the ternary nucleation parameterisation. However, Figure 10 shows that large nucleation peaks, associated with a sudden increase of $N_{10-100}$ concentrations, are erroneously simulated with the ternary nucleation parameterisation, particularly between 17-20 July.

## 6 Conclusion

This paper illustrates a method for estimating the number emission factors and the size distribution for the different activity sectors from the emissions of $PM_{2.5}$. The estimated size distribution at emissions is discretised using a sectional approach and it is refined ensuring consistency of both mass and number concentrations. This method is applied over Greater Paris and calibrated using days when nucleation was low.

$PM_1$, $PM_{2.5}$ and $PM_{10}$ concentrations are not influenced by nucleation. But nucleation has a strong influence on the number concentration in July 2009. The influence of binary nucleation, which involves sulfuric acid and water, is low over Paris. The influence of ternary nucleation, which involves sulfuric acid, ammonia and water, can be very high, but this high influence may not be realistic. It leads to very good model to measurement comparison at one suburban site, in terms of size distribution, but systematically deteriorates the correlation between simulated and measured number concentrations. Furthermore, it strongly overestimates the number concentrations at the central site, and slightly underestimate it at the other suburban site. Co-located measurements of ammonia and number concentrations are required to conclude on the role of ammonia on nucleation in

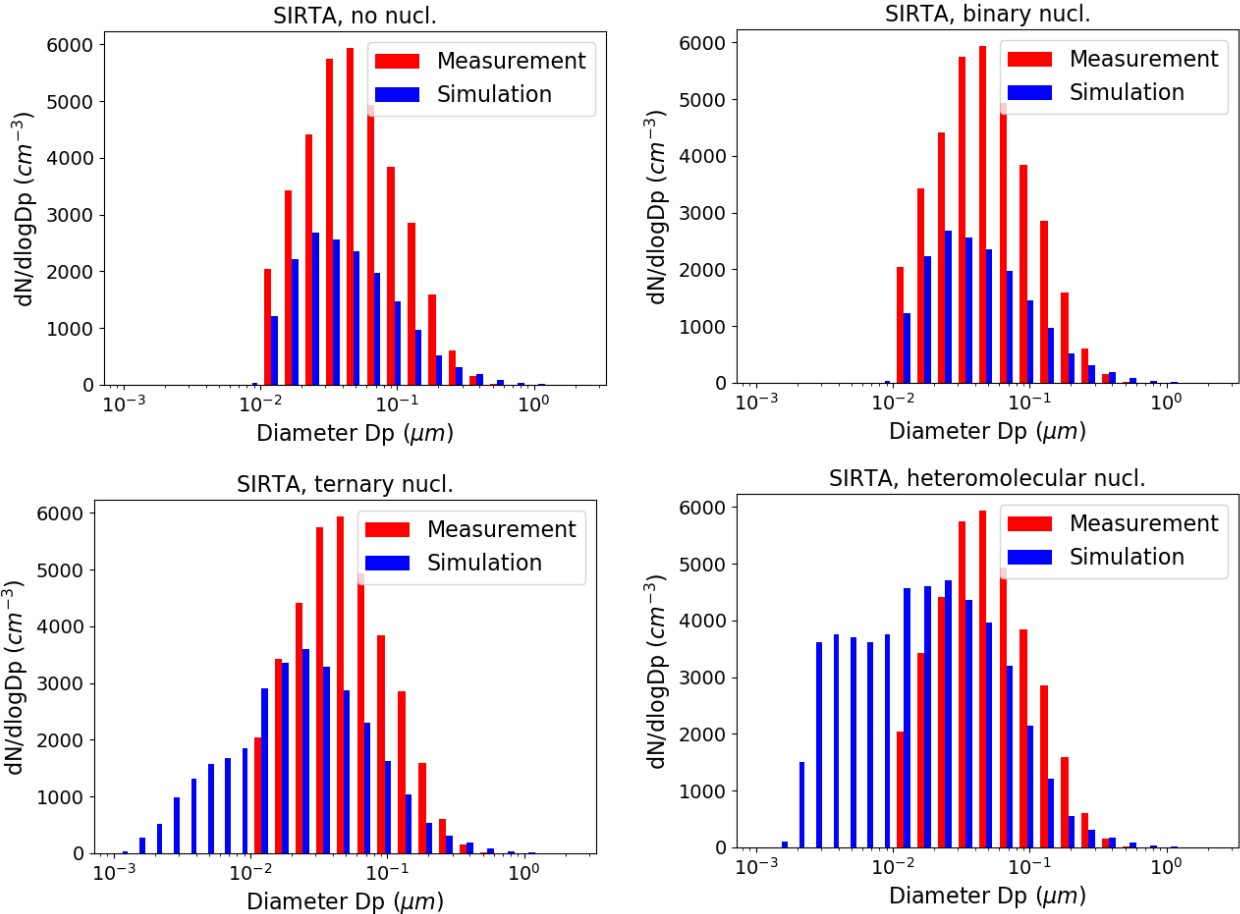

**Figure 7.** Size distribution of the number concentrations in # cm$^{-3}$ at the SIRTA site for the simulation without nucleation (top left panel), with binary nucleation (top right panel), ternary nucleation (bottom left panel), heteromolecular nucleation (bottom right panel).

urban areas. The best model to measurement comparisons for N$_{>10}$, N$_{>100}$ and the size distribution are obtained using the heteromolecular nucleation parameterisation, which involves sulfuric acid and extremely low-volatile organic compounds from monoterpenes, emphasising the realistic importance of this process.

The correlation between measured and simulated number concentrations is high for the simulation without nucleation (higher than 48%) stressing the strong influence of primary emissions and of organic vapors from traffic that influence the growth of the emitted UFP. At the two suburban sites, the heteromolecular nucleation parameterisation clearly leads to improved simulated number concentrations compared to measurements, with lower error, bias and higher correlation. The correlation is systematically deteriorated at the central site (LHVP) if nucleation is taken into account, suggesting that nucleation may involve other species than those used here, or that monoterpene emissions from traffic could be underestimated. Guo et al. (2020) suggested that extremely low-volatile compounds (ELVOC) formed from traffic emissions or emitted by traffic could

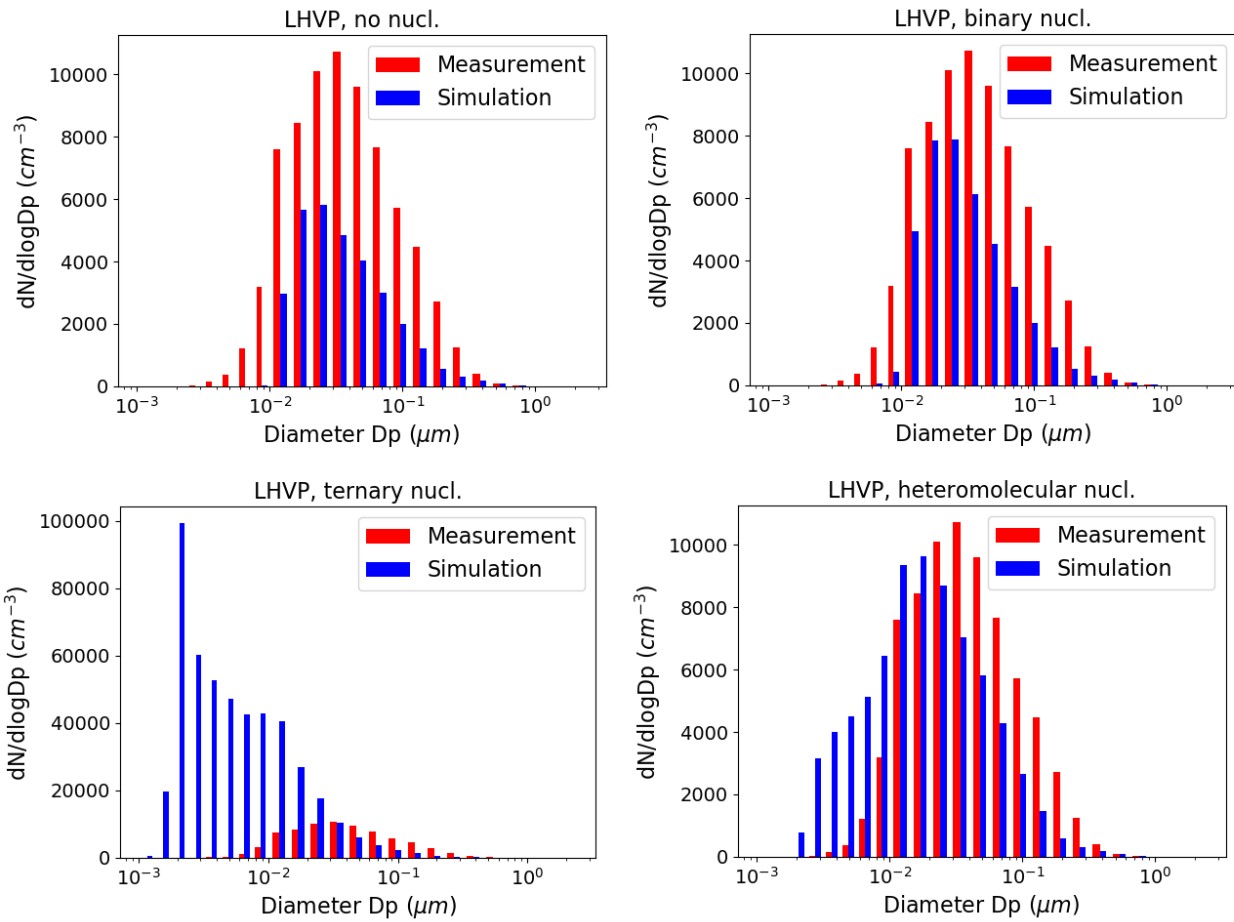

**Figure 8.** Size distribution of the number concentrations in $\# \, cm^{-3}$ at the LHVP site for the simulation without nucleation (top left panel), with binary nucleation (top right panel), ternary nucleation (bottom left panel), heteromolecular nucleation (bottom right panel).

nucleate. Recent studies suggested that traffic may emit monoterpenes, such as $\alpha$-pinene (Panopoulou et al., 2020), which could then formed ELVOC rapidly and be involved in nucleation.

## Competing interests

The authors declare that they have no conflict of interest.

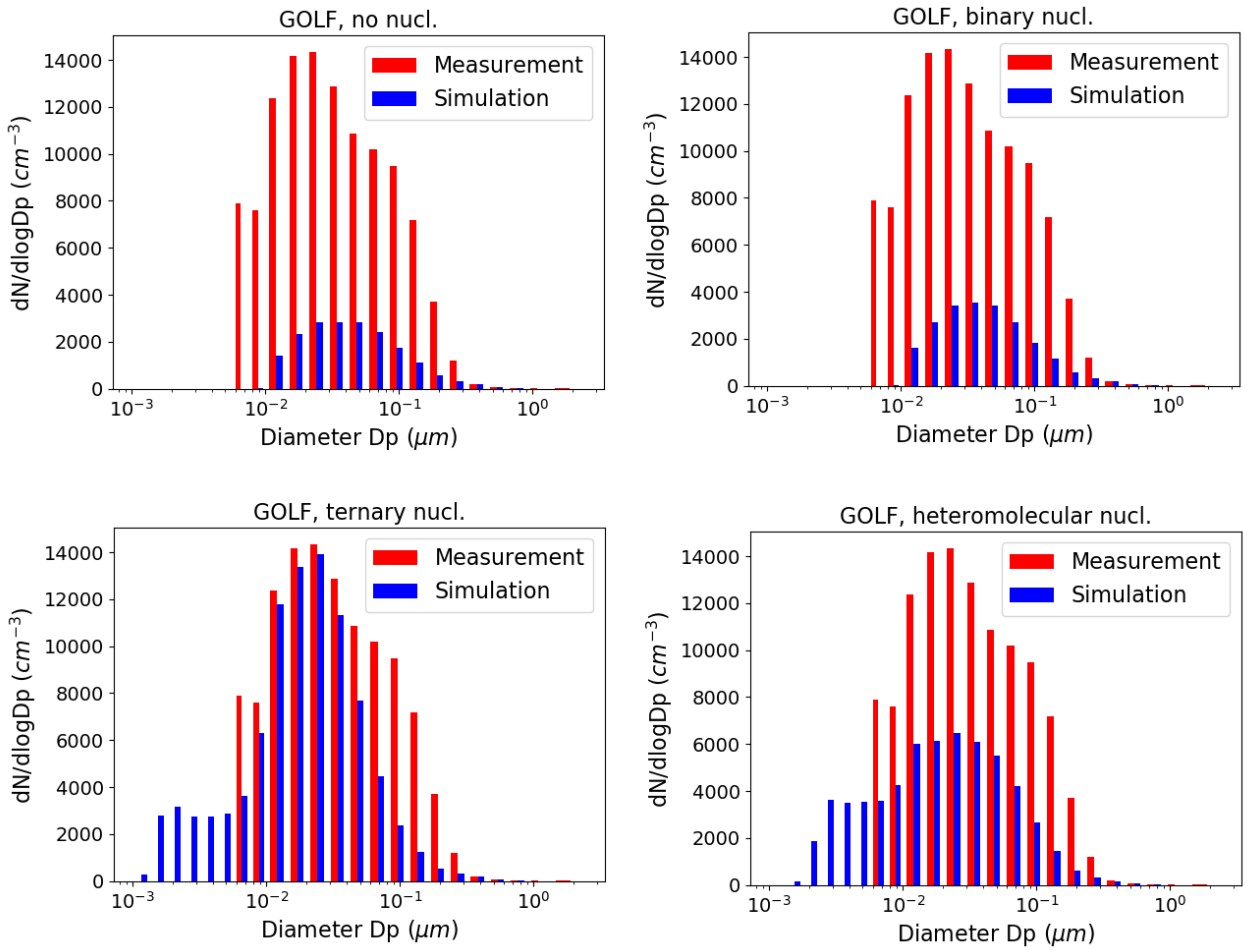

**Figure 9.** Size distribution of the number concentrations in $\# \, cm^{-3}$ at the GOLF site for the simulation without nucleation (top left panel), with binary nucleation (top right panel), ternary nucleation (bottom left panel), heteromolecular nucleation (bottom right panel).

**Acknowledgments**

The emission data used to estimate the size-distribution of particles at emissions were obtained from the web site naei.beis.gov.uk (©Crown 2021 copyright Defra & BEIS via naei.beis.gov.uk, licenced under the Open Government Licence (OGL)). This project has received funding from the European Union's Horizon 2020 research and innovation programme under grant agree-
5  ment No 101036245 (RI-URBANS). It was supported partly by ERA-PLANET (www.era-planet.eu), trans-national project SMURBS (www.smurbs.eu, grant agreement n. 689443). Thanks are due to Airparif, the Paris region air quality agency, for providing us with their air pollutant emission inventory; Evangelia Kostenidou for the SMPS measurements at the SIRTA site, and Friederike Fachinger and Johannes Schneider (Max Planck Institute for Chemistry, Germany) for the measurements at the GOLF site.

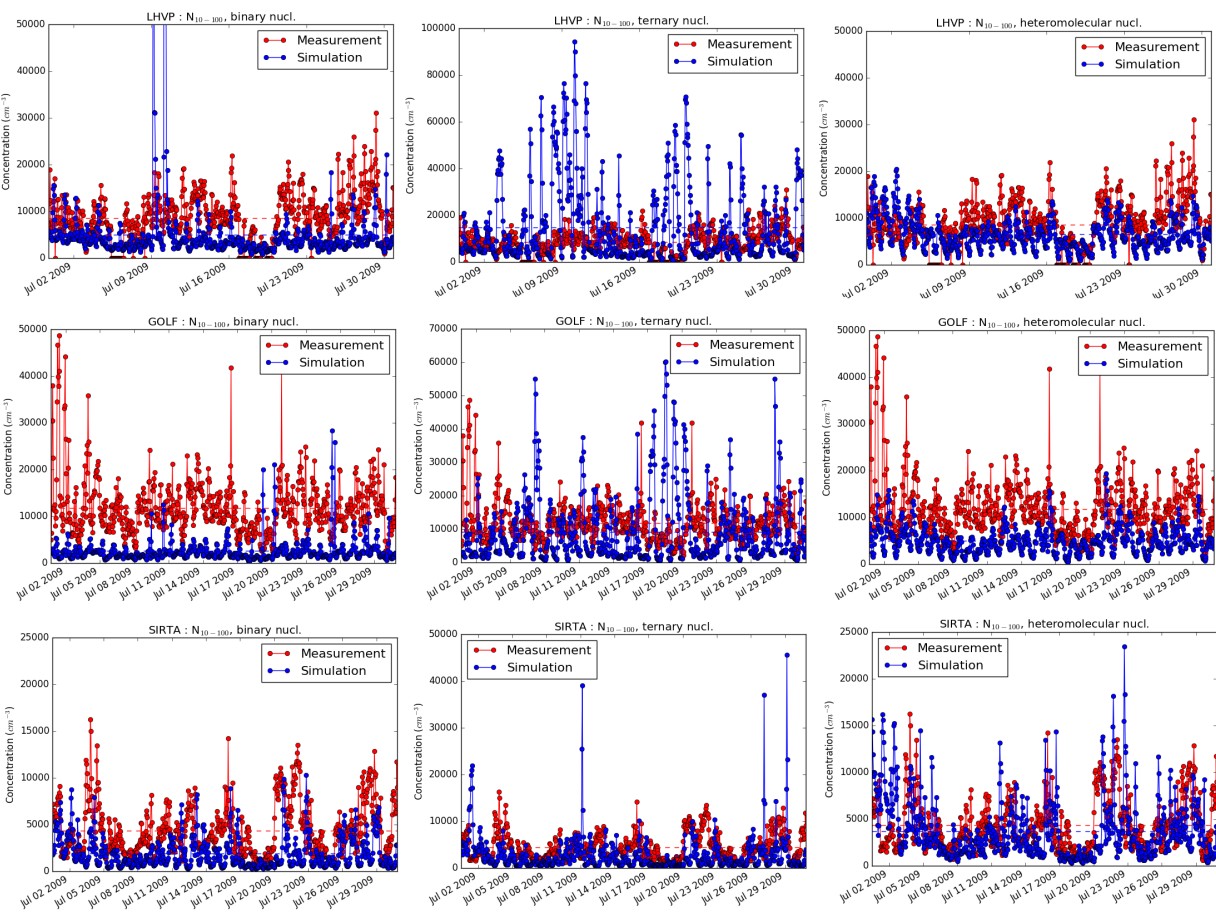

**Figure 10.** Hourly evolution of the number concentrations $N_{10-100}$ in $\# \, cm^{-3}$ at the LHVP (top horizontal panels), GOLF (middle horizontal panels) and SIRTA (lower horizontal panels) sites for the simulation with binary nucleation (left column panels), ternary nucleation (middle column panels), heteromolecular nucleation (right column panels). Note that the maximum concentration is higher in the graphs with the ternary nucleation parameterisation than in those with the binary or heteromolecular nucleation parameterisations.

## Author contributions

KS was responsible for conceptualization, conducing the visualization, validation and analysis. KS and YK developed the software. KS, YK and FC participated in the development of the aerosol model SSH-aerosol. KS, JS, MM, TP and AW acquired resources. KS wrote a first draft of the paper, which was reviewed by all co-authors.

**Code/Data availability**

The measurement data are available through the web site http://cds-espri.ipsl.upmc.fr/megapoli/index.jsp. The SSH-aerosol model can be downloaded from https://github.com/sshaerosol/ssh-aerosol. The chemistry-transport model Polair3D/Polyphemus coupled to the aerosol module SSH-aerosol and the simulated concentrations are available from the corresponding author upon
5   request.

## Appendix A: Size distribution of emissions

To estimate the size-distribution of emissions, emissions of $PM_1$, $PM_{0.1}$ are deduced from emissions of $PM_{2.5}$ using the ratios of Table A2. Then the discretisation of the size sections of the emitted particles is refined by assuming that both mass and number are kept constant in each diameter range. As an illustration, let us consider one section of bound diameters $d_-$ and

5 $d_+$, i.e. all particles in that section have diameters between $d_-$ and $d_+$ (for example, $d_- = 0.1$ $\mu$m and $d_+ = 1$ $\mu$m). These particles of diameters between $d_-$ and $d_+$ are assumed to have a mass M and a number of particles N. To refine the section discretisation into 2 sections rather than one, a bound diameter $d_m$ is added between $d_-$ and $d_+$. This allows to define two sections: section one for particles of diameters between $d_-$ and $d_m$ and section two for particles between $d_m$ and $d_+$. The particles of section 1 have diameters between $d_-$ and $d_m$, and they are assumed to have a mass $M_1$ and a number of particles

10 $N_1$. The particles of section 2 have diameters between $d_m$ and $d_+$, and they are assumed to have a mass $M_2$ and a number of particles $N_2$. We assume that refining the discretisation conserves both mass and number (M = $M_1$ + $M_2$; N = $N_1$ + $N_2$), and that the mean diameter of each section is the geometrical mean of the section. To summarize, a section of mass M and mean diameter $d_m = (d_- \, d_+)^{1/2}$ is divided into 2 sections of mass and number $(M_1, N_1)$ and $(M_2, N_2)$. The mean diameter of these sections are $\bar{d}_1 = (d_- \, d_m)^{1/2}$ and $\bar{d}_2 = (d_+ \, d_m)^{1/2}$. The conservation of number is written as

$$ N = N_1 + N_2 \tag{A1} $$

$$ \frac{M}{d_m^3} = \frac{M_1}{(d_- \, d_m)^{3/2}} + \frac{M_2}{(d_+ \, d_m)^{3/2}} \tag{A2} $$

Taking into account the conservation of mass, we have

$$ \frac{M_1 + M_2}{d_m^3} = \frac{M_1}{(d_- \, d_m)^{3/2}} + \frac{M_2}{(d_+ \, d_m)^{3/2}} \tag{A3} $$

$$ M_1 \left( \frac{1}{d_m^3} - \frac{1}{(d_- \, d_m)^{3/2}} \right) = M_2 \left( \frac{1}{(d_+ \, d_m)^{3/2}} - \frac{1}{d_m^3} \right) \tag{A4} $$

$$ M_1 \left( \frac{1}{d_m^{3/2}} - \frac{1}{d_-^{3/2}} \right) = M_2 \left( \frac{1}{d_+^{3/2}} - \frac{1}{d_m^{3/2}} \right) \tag{A5} $$

Therefore

$$ a = \frac{M_2}{M_1} = \frac{\frac{1}{d_m^{3/2}} - \frac{1}{d_-^{3/2}}}{\frac{1}{d_+^{3/2}} - \frac{1}{d_m^{3/2}}} \tag{A6} $$

Then the mass $M_1$ and $M_2$ may be written as

$$ M_1 = \frac{1}{1+a} \, M; \quad M_2 = \frac{a}{1+a} \, M \quad \text{with: } a = \frac{\frac{1}{d_m^{3/2}} - \frac{1}{d_-^{3/2}}}{\frac{1}{d_+^{3/2}} - \frac{1}{d_m^{3/2}}} $$

25 The number of particles of each section is then deduced from the mass, the density and the geometric mean diameter $\bar{d}$ of the section by assuming that particles are spherical:

$$ N_1 = \frac{6 \, M_1}{\pi \rho \, \bar{d}_1}; \quad N_2 = \frac{6 \, M_2}{\pi \rho \, \bar{d}_2} $$

Using this algorithm, the number of section is progressively refined. An example of initial diameters and the diameters after the first refinement step is shown in Table A1.

**Table A1.** Diameters (in $\mu$m) of the different sections, initially and after the first refinement step.

Initial sections

| $d_-$ | 0.01 | 0.0398 | 0.1585 | 0.631 | 2.5 |
|-------|------|--------|--------|-------|-----|
| $d_+$ | 0.0398 | 0.1585 | 0.631 | 2.5 | 10 |
| $d_m$ | 0.199 | 0.0794 | 0.316 | 1.256 | 5 |

After first refinement step

| $d_-$ | 0.01 | 0.199 | 0.0398 | 0.0794 | 0.1585 | 0.316 | 0.631 | 1.256 | 2.5 | 5 |
|-------|------|-------|--------|--------|--------|-------|-------|-------|-----|-----|
| $d_+$ | 0.199 | 0.0398 | 0.0794 | 0.1585 | 0.316 | 0.631 | 1.256 | 2.5 | 5 | 10 |
| $d_m$ | 0.0141 | 0.0282 | 0.0562 | 0.112 | 0.224 | 0.447 | 0.891 | 1.772 | 3.536 | 7.071 |

**Table A2.** Estimation of the size-distribution of emissions by estimating the ratio $PM_1/PM_{2.5}$ and $PM_{0.1}/PM_1$ for each activity sector.

| | SNAP | $PM_1/PM_{2.5}$ | $PM_{0.1}/PM_1$ |
|---|---|---|---|
| **Combustion in energy and transformation industries** | 01 | | |
| Public power and District heating plants | 0101/0102 | 0.7945 | 0.4806 |
| Petroleum refining plants | 0103 | 0.8058 | 0.5428 |
| Stationary engines | 0105 | 1 | 0.5 |
| **Non-industrial combustion plants** | 02 | | |
| Commercial and institutional plants | 0201 | 0.8577 | 0.4155 |
| Residential plants | 0202 | 0.6761 | 0.2006 |
| Plants in agriculture, forestry and aquaculture | 0203 | 0.7101 | 0.5565 |
| **Combustion in manufacturing industry** | 03 | 0.8072 | 0.5265 |
| **Production processes** | 04 | | |
| Processes in petroleum industries | 0401 | 0.8072 | 0.5265 |
| Processes in iron and steel industries and colliries | 0402 | 0.5117 | 0.3684 |
| Processes in inorganic and organic chemical industries, wood, food and other | 0404/0405/0406 | 0.7711 | 0.5 |
| **Extraction and distribution of fossil fuels and geothermal energy** | 05 | 0.8072 | 0.5265 |
| **Solvent and other product use** | 06 | 0.7711 | 0.5 |
| **Road transport** | 07 | | |
| Passenger cars, light, heavy duty vehicles and buses | 0701/0702/0703 | 0.8947 | 0.5882 |
| Mopeds and Motorcycles < 50 cm3 and > 50 cm3 | 0704/0705 | 0.6842 | 0.3846 |
| Gasoline evaporation from vehicles | 0706 | 0.8947 | 0.5882 |
| **Other mobile sources and machinery** | 08 | | |
| Railways | 0802 | 0.8947 | 0.1765 |
| Inland waterways | 0803 | 0.8975 | 0.3032 |
| Air traffic | 0805 | 0.67 | 0.2239 |
| Agriculture and Forestry | 0806/0807 | 0.8148 | 0.2575 |
| Industry | 0808 | 1 | 0.5 |
| Household and gardening | 0809 | 0.9043 | 0.5882 |
| **Waste treatment and disposal** | 09 | 1 | 0.15 |
| **Agriculture** | 10 | 1 | 0.5 |

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
