# Peer review of "Influence of emission size distribution and nucleation on number concentrations over Greater Paris"

_Atmospheric Chemistry and Physics, 2022_

## Author Comment (AC1)

*This paper presents three-dimensional simulations with a chemistry-transport model coupled to a sectional aerosol module to assess the influence of three different nucleation parameterizations on particle number concentrations and size distributions in the Greater Paris region. The authors developed an innovative methodology for distributing particulate matter emissions to the size distribution of emitted particles in the sectional representation that ensures conservation of mass and number. This method for particle emissions is successfully applied in the simulations without nucleation and with three nucleation schemes. Model results are evaluated against observations of total particle number and qualitatively also to observed size distributions at three observation sites. Overall, the manuscript is written in good style and interpretation of the results for different nucleation scheme is convincingly presented.*

*However, when comparing the modelled size distributions to measured size distributions, especially for the heteromolecular nucleation mechanism that involves biogenic oxidation products, it becomes obvious that with the current approach, the discrepancy can be attributed neither to inaccuracies of the formation rate alone nor to inaccuracies of the growth rate alone. I strongly suggest to perform a new particle formation event analysis including all nucleation-event days. The formation rate J10-50 (or J10-100) of particles can be derived from the observations (change of total number concentration plus losses through coagulation and growth) and compared to the formation rate in the model for different nucleation parametrizations. I think the paper will greatly benefit from the additional nucleation-event analysis.*

Our reply :

To compare the formation rate of the measured number concentrations with the simulated ones using the different nucleation parameterizations, a figure (Figure 10) is added to compare in July the hourly simulated and measured evolution of the number of particles of diameters between 10 and 100 µm ($N_{10-100}$). As shown in Figure 2 for the LHVP site where AIS measurements were performed, nucleation occurs through most of July after the 5th, although nucleation episodes are more scarce between 17-20 July. Therefore the hourly evolution is plotted over the whole month.

The following discussions are added in section 5.2.

« To gain more insights on the influence of the nucleation parameterisation on the size distribution and on the number formation rate, the simulated hourly evolution of the number of particles of diameters between 10 and 100 µm ($N_{10-100}$) is plotted in Figure 10 and compared to measurements. »

 « At all sites, the number concentration is strongly underestimated if only binary nucleation is taken into account. This is also shown by the hourly evolution of $N_{10-100}$ (Figure 10): the $N_{10-100}$ concentrations are under-estimated at all sites. Although a few nucleation peaks, associated with a sudden increase of $N_{10-100}$ concentrations, are simulated, they do not appear to be temporally related to those observed. »

« However, Figure 10 shows that large nucleation peaks, associated with a sudden increase of $N_{10-100}$ concentrations, are erroneously simulated with the ternary nucleation parameterisation, particularly between 17-20 July. »

« At the LHVP site, $NH_3$ concentrations vary between 0 and 3 µg m$^{-3}$ during most of July. As traffic is the main source of $NH_3$ in the center of Paris in July, the peaks of $NH_3$

concentrations are strongly related to the traffic peaks (Figure 6). When ternary nucleation is taken into account, peaks of particle number concentrations are simulated when $NH_3$ concentrations are high, and number concentrations are very low when $NH_3$ concentrations are low (Figure 10). These high peaks and variations of number concentrations are not observed in the measurements. »

« This under-estimation is less important using the heteromolecular parameterisation, which leads to an overestimation of the number of particles between 3 and 10 nm. This could be due to too low condensable vapors. The hourly variations of $N_{10-100}$ concentrations confirm that they are underestimated between 9 and 16 July and between 23 and 30 July when nucleation is the strongest at the LHVP site (Figure 2). »

*Specific Comments:*

*1.) P. 3 line 5: Another uncertainty when dealing with emissions of particulate matter is the emission of low-volatile organic vapors. Depending on the distance from the source these may be in gas phase (at high temperature) or in the condensed phase (after cooling to ambient temperature). The difficulty in accounting for the organic vapors in the emission inventory arises from the fact that they might already be partly included in PM$_{2.5}$ (as organic carbon).*

Our reply: this comment has been added to the text. In details, the sentences

« Not only the primary emissions and size distribution at emission of UFP are highly uncertain, but also their formation from gas-phase precursors (nucleation) is still not well understood. Okuljar et al. (2021) showed that sub-3 nm particles may largely be directly emitted by traffic, but this contribution may be low during nucleation episodes. »

are replaced by

« Not only the primary emissions and size distribution at emission of UFP are highly uncertain, but also their formation from gas-phase precursors (nucleation), as well as the emissions of low-volatile organic vapors, which may strongly influence the growth of UFP (Patoulias and Pandis, 2022). Okuljar et al. (2021) showed that sub-3 nm particles may largely be directly emitted by traffic, but this contribution may be low during nucleation episodes. Low-volatile organic vapors are also emitted by traffic and depending on the distance from the source, they may be in gas phase (at high temperature) or in the condensed phase (after cooling to ambient temperature). The difficulty in accounting for the organic vapors in the emission inventory arises from the fact that they might already be partly included in PM2.5 (as organic carbon, Kim et al., 2016) »

*2.) Section 2.1: Related to the previous point: how is the primary emission of semi-volatile and low-volatile organic vapors from road traffic estimated in the model?*

Our reply: The following sentences are added in the section 2.2, which deals with emissions :

« Emissions of intermediate, semi and low volatile organic compounds (IVOC, SVOC, LVOC respectively) are estimated from the organic mass of PM2.5 as detailed in Sartelet et al. (2018) : the mass of organic vapors is estimated by multiplying by 1.5 the organic

mass (Kim et al. 2016). The emitted organics are then divided into volatility classes : 25% is assigned to LVOC, 32% to SVOC and 43% to IVOC (Couvidat et al. 2012). »

*3.) Cai et al. (2018) show different influence of long-range (or short-range) transported aerosols and gaseous precursors on new particle formation events in clean and polluted environments. How is transport of particles through the regional boundaries estimated?*

Our reply: A section « Simulation setup » is added. In this section, the following sentences are added :

« The distribution of boundary conditions and emissions into 25 size sections is done offline, prior to the simulation, using the algorithm detailed in Appendix. Because the larger scale simulations from the nesting domain presented in Royer et al. (2012), Couvidat et al. (2013) did not include particles of diameters lower than 0.01 µm, the boundary conditions for particles between 0.001 and 0.01 µm are fixed to 0. »

The potential influence of transport is added in the introduction. The sentence :

« In cities, the high particle number concentrations are thought to mostly originate from nucleation and traffic emissions in summer »

is replaced by

« Although UFP may undergo transport and they may be formed elsewhere than the observation site (Cai et al. 2018), in cities, the high particle number concentrations are thought to mostly originate from nucleation and traffic emissions in summer »

*4.) P. 4, lines 17-19: The proxy of biogenic oxidation products (BioOxOrg) in the Riccobono parameterization is strictly speaking not the same as ELVOC from the auto-oxidation of monoterpenes. BioOxOrg represents later-generation oxidation products of biogenic monoterpenes. The authors should discuss what this means for the time scale of formation.*

Our reply: In the Riccobono's paper, monoterpenes are not introduced in the measurement chamber but pinanediol a first oxidation product of monoterpenes. BioOxOrg represents a myriad of oxidation products of pinanediol, and therefore BioOxOrg is refered to later-generation oxidation products. Schobesberger et al. (2013) argued that stable clusters with $H_2SO_4$ molecules may be effectively formed from highly oxidized ELVOCs. The less oxidized, but more abundant oxidation products may rather drive the initial growth of the clusters. As oxidation products of monoterpenes are also too abundant in our simulation to be considered as BioOxOrg, and referring to the observations of Schobesberger et al. (2013), we decided to use ELVOCs from the autoxidation of monoterpenes as representative of the compounds that may be involved in nucleation. Concerning the time scale, BioOxOrg is formed quickly (less than 5 min), as discussed in Schobesberger et al. (2013). This rapid formation also holds by taking BioOxOrg equal to ELVOCs.

For clarity, in section 2.1, the sentence :

« The concentration of the oxidised biogenic compounds is assumed to be equal to the concentration of extremely-low volatile organic compounds (ELVOC) formed from the autoxidation of monoterpenes (Ehn et al., 2014; Chrit et al., 2017). »

is replaced by :

« Schobesberger et al. (2013) argued that stable clusters with sulfuric acid molecules may be effectively formed from highly oxidized extremely-low volatile organic compounds (ELVOCs). The less oxidized, but more abundant oxidation products may rather drive the initial growth of the clusters. Hence, the concentration of the oxidised biogenic compounds is assumed to be equal to the concentration of ELVOCs, which are formed in the model from the autoxidation of monoterpenes (Ehn et al., 2014; Chrit et al., 2017). »

*5.) P. 4 line 23: Guo et al. (2020) find that aromatic VOC from vehicular exhaust are important precursors for nucleation and growth. While monoterpene emissions of forests may be important for regional new particle formation, I would assume that the urban local particle formation events are more likely caused by vehicle-emitted precursors.*

Our reply: Yes, we agree with the reviewer. The number of particles is underestimated in the site in central Paris, probably because of this missing source. As a parameterisation of the nucleation from precursors from vehicular exhaust is not available, this limitation is discussed in the conclusion of the paper : « Guo et al. (2020) suggested that extremely low-volatile compounds (ELVOC) formed from traffic emissions or emitted by traffic could nucleate. » Concerning the growth of UFP from the oxidation of aromatic VOCs, this is taken into account in the model, as detailed in the model description :

« The considered SOA precursors are anthropogenic (toluene, xylenes, intermediate and semi-volatile organic compounds) and biogenic (monoterpenes, sesquiterpenes, isoprene). The myriad of SOA species formed during the oxidation of those precursors is modelled with surrogate organic molecules of representative physico-chemical properties (Couvidat et al., 2012; Sartelet et al., 2020). Some of the surrogates may be considered as non-volatile: the surrogate BiAD3 (3-methyl-1,2,3-butane tricarboxylic acid) from the monoterpene oxidation, the surrogates Monomer ($C_{10}H_{14}O_9$) and Dimer ($C_{19}H_{28}O_{11}$) from the monoterpene autoxidation, the surrogate AnClP from the xylenes and toluene low-NOx oxidation, the surrogate SOAlP (secondary organic aerosol of low volatility) from the oxidation of anthropogenic semi-volatile organic compounds. The growth of UFP is strongly impacted by the condensation of low-volatility compounds as well as coagulation. »

To stress this point in the conclusion, the sentences

« The correlation between measured and simulated number concentrations is high for the simulation without nucleation (higher than 48%) stressing the strong influence of primary emissions. »

are replaced by :

« The correlation between measured and simulated number concentrations is high for the simulation without nucleation (higher than 48%) stressing the strong influence of primary emissions and of organic vapors from traffic that influence the growth of the emitted UFP. »

*6.) Section 2.2: The distribution of PM2.5 emissions over the size bins of the model is an innovative approach. It would be interesting to know if the distribution of PM2.5 emissions to size sections of emitted particles is done online in the CTM or prior to the simulation in an emission model. Further, I suggest adding a table that shows the distribution procedure for different PM size ranges to the 25 model size sections. The diameter bounds for the sections of the aerosol model could also be included in that table to avoid listing them in the text (P. 4, lines 28-29).*

Our reply: The distribution of PM2.5 emissions to size sections is done offline prior to the simulation, as now specified in section 2.2:

« The distribution of boundary conditions and emissions into 25 size sections is done offline, prior to the simulation, using the algorithm detailed in Appendix. »

More details about the demonstation of the distribution procedure are added in the appendix, as well as a Table illustrating how the diameters are refined.

*7.) P. 11, effect of the ternary nucleation: To better understand the influence of ternary nucleation in the region, it would be helpful to show the ammonia (NH3) concentration field as well and discuss the uncertainties of NH3 emissions.*

Our reply: A map of $NH_3$ concentrations over July is added, as well as the time-variation concentrations at the LHVP site. The following discussion is added to section 5.1.2 :

« No measurement of $NH_3$ concentration was available in 2009. However, the simulated concentration range is realistic, considering that measurements of $NH_3$ performed recently in the Paris city center with a mini-DOAS indicate a mean concentration of 2 µg m$^{-3}$ (Viatte et al. 2021). At the LHVP site, $NH_3$ concentrations vary between 0 and 3 µg m$^{-3}$ during most of July. As traffic is the main source of $NH_3$ in the center of Paris in July, the peaks of $NH_3$ concentrations are strongly related to the traffic peaks. When ternary nucleation is taken into account, peaks of particle number concentrations are simulated when $NH_3$ concentrations are high, and number concentrations are very low when $NH_3$ concentrations are low. These high peaks and variations of number concentrations are not observed in the measurements. »

*8.) P. 12 lines 5-7: The comparison of the size distribution from heteromolecular nucleation with observations indicates that the <20 nm particles do not grow sufficiently, either because they are too numerous or because concentrations of condensable vapor are too low. As stated in my general comments, a nucleation-event analysis would greatly help to evaluate the formation rate in the model.*

Our reply: At the SIRTA site, which is discussed in P12 lines 5-7, the heteromolecular nucleation does fairly well at representing the number of particles, but the size distribution is not perfect, because particles below 20 nm are slightly too high but overall the hourly evolution of $N_{10-100}$ shows that the evolution is well represented.

The following sentences are added : « However, Figure 10 shows that the heteromolecular nucleation allows a fairly good representation of the nucleation events associated to the different high $N_{10-100}$ concentrations. »

*Technical Corrections:*

*P. 2 line 6: should be "organs".*

Our reply : Corrected

*Figures 4 and 5: annotate longitude and latitude on the x- and y-axes of the maps.*

Our reply : Modified

*Figure 5: better denote the simulation name at the plots, for example in the header of the maps.*

Our reply : Modified

*References:*

*Cai, R., Chandra, I., Yang, D., Yao, L., Fu, Y., Li, X., Lu, Y., Luo, L., Hao, J., Ma, Y., Wang, L., Zheng, J., Seto, T., and Jiang, J.: Estimating the influence of transport on aerosol size distributions during new particle formation events, Atmos. Chem. Phys., 18, 16587-16599, https://doi.org/10.5194/acp-18-16587-2018, 2018.*

*Guo, S., Hu, M., Peng, J., Wu, Z., Zamora, M. L., Shang, D., Du, Z., Zheng, J., Fang, X., Tang, R., Wu, Y., Zeng, L., Shuai, S., Zhang, W., Wang, Y., Ji, Y., Li, Y., Zhang, A. L., Wang, W., Zhang, F., Zhao, J., Gong, X., Wang, C., Molina, M. J., and Zhang, R.: Remarkable nucleation and growth of ultrafine particles from vehicular exhaust, Proc. Nat. Acad. Sci., 117, 3427-3432, https://doi.org/10.1073/pnas.1916366117, 2020.*
***Citation***: *https://doi.org/10.5194/acp-2022-22-RC1*

Added references :

- Cai, R., Chandra, I., Yang, D., Yao, L., Fu, Y., Li, X., Lu, Y., Luo, L., Hao, J., Ma, Y., Wang, L., Zheng, J., Seto, T., and Jiang, J.: Estimating the influence of transport on aerosol size distributions during new particle formation events., Atmos. Chem. Phys., 18, 16 587–16 599, https://doi.org/10.5194/acp-18-16587-2018, 2018
- Kim, Y., Sartelet, K., Seigneur, C., Charron, A., Besombes, J.-L., Jaffrezo, J.-L., Marchand, N., and Polo, L.: Effect of measurement protocol on organic aerosol measurements of exhaust emissions from gasoline and diesel vehicles, Atmos. Environ., 140, 176–187, https://doi.org/10.1016/j.atmosenv.2016.05.045, 2016.
- Sartelet, K., Zhu, S., Moukhtar, S., André, M., André, J., Gros, V., Favez, O., Brasseur, A., and Redaelli, M.: Emission of intermediate, semi and low volatile organic compounds from traffic and their impact on secondary organic aerosol concentrations over Greater Paris, Atmos. Environ., 180, 126–137, https://doi.org/10.1016/j.atmosenv.2018.02.031, 2018.
- Schobesberger, S., Junninen, H., Bianchi, F., Lönn, G., Ehn, M., Lehtipalo, K., Dommen, J., Ehrhart, S., Ortega, I. K., Franchin, A., Nieminen, T., Riccobono, F., Hutterli, M., Duplissy, J., Almeida, J., Amorim, A., Breitenlechner, M., Downard, A. J., Dunne, E. M., Flagan, R. C., Kajos, M., Keskinen, H., Kirkby, J., Kupc, A., Kürten, A., Kurtén, T., Laaksonen, A., Mathot, S., Onnela, A., Praplan, A. P., Rondo, L., Santos, F. D., Schallhart, S., Schnitzhofer, R., Sipilä, M., Tomé, A., Tsagkogeorgas, G., Vehkamäki, H., Wimmer, D., Baltensperger, U., Carslaw, K. S., Curtius, J.,

Hansel, A., Petäjä, T., Kulmala, M., Donahue, N. M., and Worsnop, D. R.: Molecular understanding of atmospheric particle formation from sulfuric acid and large oxidized organic molecules, Proc. Nat. Acad. Sci., 110, 17 223–17 228, https://doi.org/10.1073/pnas.1306973110, 2013.

- Viatte, C., Petit, J.-E., Yamanouchi, S., Van Damme, M., Doucerain, C., Germain-Piaulenne, E., Gros, V., Favez, O., Clarisse, L., Coheur,P.-F., Strong, K., and Clerbaux, C.: Ammonia and PM2:5 Air Pollution in Paris during the 2020 COVID Lockdown, Atmosphere, 12, https://doi.org/10.3390/atmos12020160, 2021

---

## Author Comment (AC2)

*Sartelet and co-authors present a comparison of modelled and measured aerosol number concentrations and size distributions at three sites in the greater Paris region. Modelled aerosol concentrations are performed without a nucleation scheme and using three nucleation schemes: binary, ternary, and heteromolecuar (including organic components). The paper is clearly written and the analysis of data is mainly about comparing correlation model results and measurements. My expertise is not in aerosol modelling, so I cannot comment on the validity of the modelling framework, however I have a few comments pertaining to the treatment of aerosol emissions that are shown below. In summary, I believe that there is a large number of processes that can influence the aerosol number concentrations (especially the nucleation mode) that are not addressed nor discussed in this study.*

*It is not clear to me how representative these modelling simulations are compared to measurements given that there are significant uncertainties regarding emissions of gas-phase precursors, especially organic SOA precursors. Have the authors investigated the impacts of changing precursor source emission concentrations and evaluating the impacts on number concentrations and size distributions? How would that affect model results?*

Our reply: Emissions and formation of gas-phase precursors of particles are indeed attached to significant uncertainties. To check that our simulation is realistic, the concentrations of $PM_1$ directly related to these precursors are compared to measurements. Nitrate, ammonium, sulfate and organics compare very well to the measurements, as detailed in section 5.1.1., as well as gas-phase $NO_2$. For $NH_3$, the following sentences are added :

« No measurement of $NH_3$ concentration was available in 2009. However, the simulated concentration range is realistic, considering that measurements of $NH_3$ performed recently in the Paris city center with a mini-DOAS indicate a mean concentration of 2 µg m$^{-3}$ (Viatte et al. 2021). »

*Do the authors consider the emissions of low- and extremely-low organic volatile compounds? if so, how are these treated? if not, I would assume that these LVOCs and ELVOCs can significantly impact modelled nucleation rate, concentrations, and size distributions, hence a discussion of potential impacts would be needed in the manuscript.*

Our reply: Emissions of volatile compounds of different volatilities are treated in the model. The following sentences are added in the section 2.2, which deals with emissions :

« Emissions of intermediate, semi and low volatile organic compounds (IVOC, SVOC, LVOC respectively) are estimated from the organic mass of PM2.5 as detailed in Sartelet et al. (2018) : the mass of organic vapors is estimated by multiplying by 1.5 the organic mass (Kim et al. 2016). The emitted organics are then divided into volatility classes : 25% is assigned to LVOC, 32% to SVOC and 43% to IVOC (Couvidat et al. 2012). »

The ELVOCs involved in the heteromolecular nucleation are not emitted in the atmosphere but formed from autoxidation of monoterpenes (Ehn et al 2014).

*How are organics partitioned to the aerosol phase? Different treatments can affect the evaporation of the nucleation mode (and hence concentrations and size distribution), especially when using the heteromolecular nucleation scheme which consider condensation of organic components. For instance, have the authors explored the impact of absorptive partitioning versus non-adsorptive partitioning? I am bringing this issue since*

*it will likely affect nucleation mode number concentrations, which is the main focus of this study.*

Our reply: Organics partition to the aerosol phase either by absorbing into an organic phase or into an aqueous phase, depending on the properties of the compounds (hydrophilic or hydrophobic). Adsorption is not considered, as we assume that it is negligible in front of absorption because there is always aerosol mass in the atmosphere, and non-volatile compounds condense independently of the particle composition. As detailed in section 2.1, the growth of UFP is strongly impacted by the condensation of low-volatility compounds as well as coagulation. Therefore, numerically, the condensation of non-volatile compounds is solved dynamically with nucleation and coagulation processes. Those non-volatile compounds, such as the ELVOCs involved in the the heteromolecular nucleation, may not evaporate from the nucleation mode, because they are not volatile. Then the condensation/evaporation of semi-volatile compounds is computed by assuming bulk thermodynamic equilibrium between the gas and the particle phases.

*Can the author add a temporal comparison of modelled and measured $N_{<10}$, $N_{<100}$ and $N_{>100}$ for the duration of the measurements? It would be informative to understand the temporal evolution of particle concentrations and chech if biases occur on given days or are consistent throughout the measurement period.*

Our reply: The hourly temporal evolution of $N_{10-100}$ is added and compared to measurements for the different nucleation parameterisations in section 5.2. $N_{10-100}$ concentrations are plotted because they allow us to clearly visualize the strengths and shortcomings of the different nucleation parameterisations for the formation of UFP, and to perform coherent comparisons at all sites.

Added references :

- Cai, R., Chandra, I., Yang, D., Yao, L., Fu, Y., Li, X., Lu, Y., Luo, L., Hao, J., Ma, Y., Wang, L., Zheng, J., Seto, T., and Jiang, J.: Estimating the influence of transport on aerosol size distributions during new particle formation events., Atmos. Chem. Phys., 18, 16 587–16 599, https://doi.org/10.5194/acp-18-16587-2018, 2018
- Kim, Y., Sartelet, K., Seigneur, C., Charron, A., Besombes, J.-L., Jaffrezo, J.-L., Marchand, N., and Polo, L.: Effect of measurement protocol on organic aerosol measurements of exhaust emissions from gasoline and diesel vehicles, Atmos. Environ., 140, 176–187, https://doi.org/10.1016/j.atmosenv.2016.05.045, 2016.
- Sartelet, K., Zhu, S., Moukhtar, S., André, M., André, J., Gros, V., Favez, O., Brasseur, A., and Redaelli, M.: Emission of intermediate, semi and low volatile organic compounds from traffic and their impact on secondary organic aerosol concentrations over Greater Paris, Atmos. Environ., 180, 126–137, https://doi.org/10.1016/j.atmosenv.2018.02.031, 2018.
- Schobesberger, S., Junninen, H., Bianchi, F., Lönn, G., Ehn, M., Lehtipalo, K., Dommen, J., Ehrhart, S., Ortega, I. K., Franchin, A., Nieminen, T., Riccobono, F., Hutterli, M., Duplissy, J., Almeida, J., Amorim, A., Breitenlechner, M., Downard, A. J., Dunne, E. M., Flagan, R. C., Kajos, M., Keskinen, H., Kirkby, J., Kupc, A., Kürten, A., Kurtén, T., Laaksonen, A., Mathot, S., Onnela, A., Praplan, A. P., Rondo, L., Santos, F. D., Schallhart, S., Schnitzhofer, R., Sipilä, M., Tomé, A., Tsagkogeorgas, G., Vehkamäki, H., Wimmer, D., Baltensperger, U., Carslaw, K. S., Curtius, J., Hansel, A., Petäjä, T., Kulmala, M., Donahue, N. M., and Worsnop, D. R.: Molecular

understanding of atmospheric particle formation from sulfuric acid and large oxidized organic molecules, Proc. Nat. Acad. Sci., 110, 17 223–17 228, https://doi.org/10.1073/pnas.1306973110, 2013.

- Viatte, C., Petit, J.-E., Yamanouchi, S., Van Damme, M., Doucerain, C., Germain-Piaulenne, E., Gros, V., Favez, O., Clarisse, L., Coheur,P.-F., Strong, K., and Clerbaux, C.: Ammonia and PM2:5 Air Pollution in Paris during the 2020 COVID Lockdown, Atmosphere, 12, https://doi.org/10.3390/atmos12020160, 2021

---

## Author Response (AR2)

*Comments from reviewer 1 : The method to redistribute number concentrations in the low diameter range of emissions uses a fitting parameter, as stated in the Abstract. In sections 2.3 and 3, this parameter is called distribution coefficient alpha_em. In section 2.3 it should be better explained how the pre-defined number size distribution has been obtained at emission, and how the parameter is applied to modulate the pre-defined size distribution. The parameter value is finally deduced in a sensitivity analysis for a non-event period by comparing simulated number concentrations against measured data. If a fitting procedure was involved, the goodness of fit for the different variations of the parameter should be provided. Otherwise, I suggest to use the same name for the parameter in the abstract as in the manuscript* text.

Our reply : No fitting procedure was involved, therefore the same name was added for the parameter in the abstract.

In section 2.3, the sentences : « However, the mass of particles PM0.1 is redistributed arbitrarily between the low range of diameters (between 0.01 µm and 0.0398 µm) and the high range (above 0.0398 µm) using a distribution coefficient $\alpha_{em}$ » are replaced by

 « However, the mass of UFP particles PM0.1 is redistributed arbitrarily between the low range of UFP diameters (between 0.01 µm and 0.0398 µm) and the high range (above 0.0398 µm) using a distribution coefficient $\alpha_{em}$, i.e. an emission ratio ($\alpha_{em}$, (1- $\alpha_{em}$ )) distributes PM0.1 in 2 size ranges. »

Furthermore the sentences « The choice of this arbitrary distribution coefficient, and a sensitivity study to it, is performed in section 3. » are replaced by

« The mass allocated to the section between 0.01 µm and 0.0398 µm corresponds to $\alpha_{em}$  times the mass of $PM_{0.1}$, and the mass allocated to the section between 0.0398 µm and 0.1 µm corresponds to (1- $\alpha_{em}$ ) times the mass of $PM_{0.1}$. To determine this arbitrary distribution coefficient, simulations are compared to measurements during non-nucleation days using three different value of $\alpha_{em}$ : 10%, 15% and 25% (section 3) »